# Genome-Wide Association Study (GWAS) for resistance to *Sclerotinia sclerotiorum* in Common Bean

**DOI:** 10.3390/genes11121496

**Published:** 2020-12-12

**Authors:** Ana Campa, Carmen García-Fernández, Juan José Ferreira

**Affiliations:** Plant Genetics Group, Regional Service for Agrofood Research and Development (SERIDA), Ctra AS-267 PK 19, 33300 Asturias, Spain; cgarcia@serida.org (C.G.-F.); jjferreira@serida.org (J.J.F.)

**Keywords:** *Phaseolus vulgaris* L., common bean, white mold, *Sclerotinia sclerotiorum*, diversity panel, genome-wide association study, resistance genes

## Abstract

White mold (WM) is a devastating fungal disease affecting common bean (*Phaseolus vulgaris* L.). In this research, a genome-wide association study (GWAS) for WM resistance was conducted using 294 lines of the Spanish diversity panel. One single-locus method and six multi-locus methods were used in the GWAS. Response to this fungus showed a continuous distribution, and 28 lines were identified as potential resistance sources, including lines of Andean and Mesoamerican origin, as well as intermediate lines between the two gene pools. Twenty-two significant associations were identified, which were organized into 15 quantitative trait intervals (QTIs) located on chromosomes Pv01, Pv02, Pv03, Pv04, Pv08, and Pv09. Seven of these QTIs were identified for the first time, whereas eight corresponded to chromosome regions previously identified in the WM resistance. In all, 468 genes were annotated in these regions, 61 of which were proposed potential candidate genes for WM resistance, based on their function related to the three main defense stages on the host: recognition (22), signal transduction (8), and defense response (31). Results obtained from this work will contribute to a better understanding of the complex quantitative resistance to WM in common bean and reveal information of significance for future breeding programs.

## 1. Introduction

Common bean, *Phaseolus vulgaris* L., is the most important grain legume for human consumption worldwide [1] and can be harvested fresh as snap beans (pods harvested before the seed development phase) or as dry beans (seeds harvested at complete maturity). It is a diploid species (2n = 2x = 22) native to the Americas, in which two major eco-geographically and genetically distinct gene pools, the Andean (AN) and the Mesoamerican (MA) have been described [2,3,4,5,6]. Common bean was introduced into Europe from the Americas in the 16th century, and since then, a considerable diversity of this crop has been traditionally grown in many Spanish regions. Part of this traditional Spanish diversity has been gathered in the Spanish Diversity Panel SDP [7].

Diseases are one of the main constraint factors in common bean production. Among them, white mold is responsible for important losses of bean yield production on a global scale. The causal agent of white mold in the common bean is the fungus *Sclerotinia sclerotiorum* (Lib.) de Bary, which is capable of infecting over 400 plant species worldwide [8,9]. Symptoms of the disease are easily identified with a white cottony appearance caused by the fungal mycelium on stems, leaves, flowers, or pods. Resting structures called sclerotia are produced during the infection process as hardened, dense mycelial bodies, that can survive in the soil for several years, increasing the incidence of the disease when susceptible varieties are grown. The wide host range and prolonged survival in the soil make conventional control measures against *S. sclerotiorum* more problematic, increasing the importance of developing resistant bean varieties. The fungus exhibits a sophisticated pathogenic process, showing an initial biotrophic lifestyle, followed by a subsequent necrotrophic lifestyle [10]. Recently, a model for the plant defense process against *S. sclerotiorum* infection has been proposed based on the current knowledge from different dicotyledonous hosts [11]. This plant defense model can be split into three main stages: (i) recognition of pathogen-associated molecules at the cell wall level, involving proteins for recognition, such as nucleotide-binding site-leucine-rich repeats (NBS-LRR), and different pathways, such as calcium signaling; (ii) signal-transduction at the cytoplasmic level, involving kinase cascades or signal molecules like salicylic acid, jasmonic acid, ethylene, auxin, abscisic acid, or nitric oxide, as well as reactive oxygen species; (iii) activation of specific defense responses, including transcription factors, protein kinases, pathogenesis-related proteins, reactive oxygen species production, oxidative protection, callose deposition, and production of camalexin and other specialized secondary plant metabolites [11].

Defense against this pathogen in the common bean is conditioned by both avoidance and physiological resistance mechanisms, both being quantitatively inherited [12]. Since 2001, several quantitative trait loci (QTLs) have been mapped for WM resistance and disease avoidance using different bi-parental mapping populations [12,13,14,15,16,17,18,19,20,21,22,23]. Due to the availability of an annotated reference genome sequence in common bean [24], it has been possible to define the physical position of several QTLs mapped in different populations and to establish a correspondence between them. Vasconcellos et al. [23] defined the physical position of 37 QTLs, which were condensed into 17 loci, of which nine were identified as consensus genomic regions, called meta-QTLs. However, bi-parental QTL mapping populations show some limitations, because only allelic diversity that segregates between the parents can be detected, or the limited number of recombination events decreases the mapping resolution [25].

A complementary strategy for the detection of QTLs is the use of diversity panels and genome-wide association studies (GWAS), which overcome the main limitations listed above for bi-parental mapping populations. Different methods can be used for conducting a GWAS, most being based on single-locus one-dimensional models, which do not provide true information about complex traits that are frequently controlled by multiple loci showing epistatic relationships. The single-locus model for GWAS used conventionally is the mixed linear model MLM [26,27]. Following the establishment of the MLM for single-locus analysis, several multi-locus-based methods have been proposed. The first was the multi-locus random-SNP-effect mixed linear model mrMLM [28], followed by pLARmEB [29], ISIS EM-BLASSO [30], FASTmrMLM [31], FASTmrEMMA [32], and pKWmEB [33]. To date, only one preliminary GWAS for WM resistance has been conducted in common bean, using the MLM and two diversity panels, namely the Snap Bean Diversity Panel of 150 cultivars and breeding lines, and the Snap Bean Association Panel, consisting of 382 cultivars and breeding lines [34]. This preliminary work described 39 chromosome regions involved in WM resistance, 25 of which were newly identified, with the remaining 14 corresponding to regions identified in other studies.

In the present study, a GWAS for WM resistance was conducted considering seven different GWAS methods, based on single-locus and multi-locus analyses, and using the SDP, a panel including wide phenotypic and genotypic diversity. Results will broaden the understanding of the genetic architecture of this complex quantitative resistance trait in common bean.

## 2. Materials and Methods

### 2.1. Plant Material

A total of 294 lines from the SDP were used in this study [7]. This panel was established from local Spanish germplasm, as well as including old and elite cultivars, mainly used for snap consumption. Most of the landraces included were derived from the Spanish common bean core collection [35]. According to Pascual et al. [36], lines Cornell49242 and AB136 were used as checks of susceptibility and high resistance responses, respectively.

Genotyping-by-sequencing GBS, [37] was carried out at BGI-Tech (Copenhagen, Denmark) using the *ApeKI* restriction enzyme. The sequencing reads obtained were aligned using the *Phaseolus vulgaris* L. genome v1 (www.ncbi.nlm.nih.gov; [24]) and a total of 9070 single-nucleotide polymorphism (SNPs) were mapped [7]. Data were filtered in Tassel v 5.2 software [38] for missing values (<10%) and minor allele frequencies (MAF ≥ 0.05), being considered for GWAS 5394 SNPs distributed across the eleven bean chromosomes (Appendix A). Each SNP was named to indicate the chromosome onto which it was mapped, followed by its physical location in base pairs.

### 2.2. White Mold Response Evaluation

Resistance tests were conducted using the straw method under controlled greenhouse conditions [39]. All lines were evaluated in two separate experiments (E1: sowing day 17 of April 2018; E2: sowing date 26 of April 2018). Each experiment consisted of two replicates (pots) per line, with 4–5 plants per pot, arranged in a randomized complete block design. Seeds were sown in 15 cm diameter plastic pots of 1.5 L volume containing 80% peats: 20% perlite and were watered and fertilized for normal plant growth. Each plant was inoculated when the third internode of the main stem was fully developed (21–28 days after planting).

The local *S. sclerotiorum* isolate, WM2, derived from a single hyphal tip and conserved in the form of sclerotia at −20 °C, was used for the resistance test [36]. One sclerotium was germinated on potato dextrose agar medium (Difco, Becton, Dickinson and Company, Spark, MD, USA) in a standard Petri plate and then subcultured to prepare plates of fresh and actively growing mycelium for the inoculations. A single mycelial plug was placed, mycelium-side down, on the cut main stem. The inoculated plants were maintained at moderate temperature (16 °C to 24 °C) in the shade and at a high relative humidity (70%). Disease progression was evaluated 8–10 days after inoculation (when the susceptible control was dead), based on the level of invasion of the main stem, using a 1–9 severity scale, where 1 = no symptoms and 9 = invasion of the third node and total plant collapse [40]. Values equal to or less than 4.5 were considered to be a resistant reaction (R), values between 4.5 and 7 were considered to be an intermediate reaction (I), and values equal to or greater than 7 were considered to be susceptible (S). Each plant in the pot was rated separately. The disease value per pot was calculated as the arithmetic mean of the 4–5 plants. Mean values were adjusted identifying outliers through the coefficient of variation (CV = (standard deviation/mean) × 100). Coefficient of variation over 50% was not accepted. In each evaluation, E1 and E2, the disease value per genotype was calculated as the average of the two replicates (pots). E1, E2, and the arithmetic mean of E1 and E2 (Em) were considered to be three sets of data.

### 2.3. Data Analysis

Statistical analyses of the genotypic and phenotypic data were conducted with the R Project for Statistical Computing [41]. The distribution of the SNPs along the eleven chromosomes was visualized with the “CMplot” package in R. Spearman’s correlation coefficient was calculated to estimate the association between results from the independent inoculations, E1 and E2.

Broad-sense heritability (*H*^2^) for the trait “WM response” within the 294 SDP lines was estimated using the function repeatability() of the “heritability” package in R. *H*^2^ was estimated at plot level in the E1 and E2 evaluations following the equation *V_g_*/(*V_g_* + *V_e_*), where *V_g_* = (*MS*(*G*) − *MS*(*E*)/*r* and *V_e_* = *MS*(*E*), *r* is the number of replicates per genotype, and *MS*(*G*) an *MS*(*E*) are the mean sums of squares for genotype and residual error obtained from the analysis of variance. *H*^2^ was estimated at the genotype level in the Em data set following the equation *V_g_*/(*V_g_* + *V_e/_r*).

### 2.4. Genome-Wide Association Study

To dissect the genetic architecture of a trait as complex as white mold resistance, two approaches were used in this study: (i) consistency between analysis in different data sets (E1, E2, and Em) and (ii) multiple GWAS methodologies and models.

Two different GWAS methodologies were considered: single-locus GWAS (SL-GWAS) based on MLM [26,27] and multi-locus GWAS (ML-GWAS) based on six models: mrMLM, FASTmrMLM, FASTmrEMMA, ISIS EM-BLASSO, pKWmEB, and pLARmEB [28−33]. SL-GWAS was conducted in Tassel v 5.2 [38], while ML-GWAS was performed using the “mrMLM” package [42] in R. To account for the multiple levels of relatedness within the lines included in the panel, population structure and kinship matrix were considered. Population structure was measured by Principal Component Analysis (PCA) conducted on the genotypic data set, considering two principal components according to the results of Campa et al. [7]. The centered-IBS method was used to obtain the kinship matrix, which is an estimate of the additive genetic variance. Both, PCA and kinship matrices were obtained in Tassel v5.2 [7] and used in all the GWAS models. A threshold of −log10(*p*) equal to or greater than 3.0 was used to identify significant associations between a trait and an SNP.

Three data sets (E1, E2, and Em) were evaluated for each one of the seven GWAS methods (MLM, mrMLM, FASTmrMLM, FASTmrEMMA, ISIS EM-BLASSO, pKWmEB, and pLARmEB). For each method, only significant associations identified in the mean data set (Em) and in, at least, one experimental environment (E1, E2) were considered. Each quantitative trait nucleotide (QTN) was named based on the chromosome and physical position in Mbp of the associated SNP (i.e., the SNP marker located at 42.46 Mbp of chromosome Pv01 would result in the WM1_42.46).

To verify the robustness of these QTNs, T-student tests were conducted to detect significant differences for the response to WM between the two alleles of the corresponding SNP. The observed variation was visualized in a boxplot.

### 2.5. Quantitative Trait Intervals

Quantitative trait intervals (QTIs) were defined as a group of QTNs located at less than 100-kbp upstream and downstream [43] from the significant SNP. In some cases, the physical positions of different QTIs overlapped or were situated very close to one another. Physical positions less than 400-kbp apart were considered to be overlapping QTIs. QTIs were named using the nomenclature chromosome and the lowest physical position in Mbp of the QTI (i.e., if the QTI expands from 42.46 to 43.02 Mbp of chromosome Pv01, it would be named as QTI1_42).

The “ShinyCircos” package [44] in R was used to visualize the position of each QTI and meta-QTLs/QTLs/QTNs previously reported in the bean genome from the underlying markers [23,34].

### 2.6. Candidate Gene Identification

Genes underlying each QTI were analyzed using the legume information system (https://legumeinfo.org/). Potential candidate genes were considered, based on their function and according to the three main stages proposed in the defense process against *S. sclerotiorum*: recognition, signal transduction, and defense response [11]. Genes coding for pathogenesis-related proteins or stress-antifungal proteins were also considered as potential candidate genes.

## 3. Results

### 3.1. White Mold Evaluation

A total of 294 common bean lines included in the SDP were evaluated against the local WM isolate in two independent evaluations, E1 and E2. E1 and E2 were significantly positively correlated (*R* = 0.68, *p* < 0.01; Appendix A). Figure 1 shows the distribution of the reaction to inoculation observed over the 1 to 9 disease severity scale. A wide and continuous distribution of disease reactions was observed. The broad-sense heritability of the trait WM response was estimated to be *H*^2^ = 0.69 in the E1, *H*^2^ = 0.78 in the E2, and *H*^2^ = 0.57 in the Em.

Most lines showed an intermediate or susceptible reaction on the disease severity scale. A total of 28 lines showed values less than 4.5 and were potential resistance sources against WM (Table 1). According to Campa et al. [7], seven of these lines came from the Andean gene pool of common bean, seven from the Mesoamerican, and 14 were recombinants between the two gene pools. Most of these potential resistance sources exhibited an indeterminate growth habit [7].

### 3.2. Genome-Wide Association Study

For SL-GWAS, 55 significant WM-SNP associations were identified, using the three WM data sets independently (Appendix A): 15 WM-SNP associations were identified for E1, 23 for E2, and 17 for Em. Of these associations, 13 were considered to be robust QTNs located on chromosomes Pv01, Pv02, Pv03, Pv04, Pv08, and Pv09 (Table 2).

For ML-GWAS, 147 significant WM-SNP associations were identified in the three WM data sets (Appendix A). The mrMLM method reveals 34 significant associations, of which only three were consistent QTNs located on chromosomes Pv01, Pv02, and Pv09 (Table 2). For the FASTmrMLM method, 20 significant associations were identified, three of which were considered to be QTNs located on chromosomes Pv01, Pv03, and Pv08. For the FASTmrEMMA, 33 significant associations resulted in the identification of three QTNs on chromosomes Pv01 and Pv08. For the ISIS EM-BLASSO, 22 significant associations resulted in four QTNs on chromosomes Pv01, Pv04, and Pv08. For the pKWmEB, 27 significant associations resulted in three QTNs on chromosomes Pv04 and Pv08. Finally, for the pLARmEB 33 significant associations were identified, giving rise to eight QTNs located on Pv01, Pv02, Pv03, Pv04, Pv08, and Pv09.

In sum, 22 robust QTNs, located on chromosomes Pv01, Pv02, Pv03, Pv04, Pv08, and Pv09, were obtained from SL-GWAS and ML-GWAS and the three data sets considered (Table 2). Among them, five QTNs were identified using more than one GWAS method. Special mention should be made for WM1_45.59 and WM8_52.39, which were identified by seven and six GWAS methods, respectively. To verify the robustness of these 22 QTNs, Student’s t-tests were carried out for the mean WM evaluation (Figure 2). Significant differences were observed in all cases.

### 3.3. Quantitative Trait Intervals

The 22 QTNs identified were grouped into 15 QTIs (Table 3). Three QTIs were located in a telomeric position of chromosome Pv01, two QTIs were on Pv02, two on Pv03, one on Pv04, four on Pv08, and three on Pv09. The physical positions of these QTIs were compared with those of WM resistance loci previously reported in common bean. Figure 3 shows the overlapping between chromosome regions identified in this work and the QTLs, meta-QTLs, or QTNs identified in previous studies [23,34]. In all, seven new QTIs were identified in the current study as regions involved in the response against WM: QTI1_42, QTI2_34, QTI3_0.9, QTI4_2, QTI8_52, QTI8_55, and QTI9_36. The remaining eight QTIs identified corresponded to regions previously associated with WM resistance: QTI1_45, QTI1_50, QTI2_4, QTI3_39, QTI8_13, QTI8_37, QTI9_11, and QTI9_16.

### 3.4. Candidate Genes

Genes underlying each QTI (spanning 5.5 Mbp) were investigated to identify potential candidate genes involved in the WM response. There were 468 genes annotated, 61 of which were identified as potential candidate genes based on their function (Table 3; Appendix A). Twenty-two proteins were related to the recognition of the pathogen, eight were related to signal transduction, and 31 were related to defense response. No candidate genes were identified at QTI2_34, QTI8_13, and QTI8_37.

## 4. Discussion

The use of diversity panels has notably increased since the development of advanced genomic sequencing technologies and the availability of annotated reference genome sequences. In the current work, the resistance response against a local isolate of *S. sclerotiorum* was studied through GWAS, using the common bean SDP. The SDP was established from local Spanish germplasm, including the Spanish common bean core collection [35], and is considered to be representative of the Spanish genetic diversity for this species [7]. Using the straw test evaluation under greenhouse conditions, a moderate broad-sense heritability was observed for WM resistance. Heritability estimated for WM resistance in previous studies was also low to moderate [12,13,45,46], indicating the large environmental effect in the expression of this resistance.

A total of 28 SDP lines were identified as potential resistance sources against WM, most of them having indeterminate growth habit. Nine of these lines were obtained from the commercial snap varieties Sacha, Bilma, Helda, Planeta, Donna, Vitalis, Florencia, Marconi, and Tendergreen [7] so it could not be excluded that these resistances derived from different breeding programs in commercial varieties. The remaining 19 resistance sources were obtained from local Spanish landraces, which were assigned to different gene pools of common beans: six of them were genetically close to the Andean gene pool, six to the Mesoamerican gene pool and seven were recombinants between the two gene pools [7]. Most of the WM resistance sources identified to date are of Andean origin or from the secondary gene pool such as *Phaseolus coccineus* [23] so that the identification of six putative new resistance sources from the Mesoamerican gene pool is a significant finding.

Lines SDP035 and SDP071, identified as resistant in the current study, are derived from the Spanish landraces BGE003121 and BGE04000, respectively, and are both included in the Spanish common bean core collection [7,35]. This result agrees with that of Pascual et al. [36] who identified these Spanish landraces as potential sources of resistance against WM. The accession BGE003121 was identified as being of Andean origin based on 12 molecular markers, including the seed protein phaseolin [35]. Later, the use of 3099 SNP markers, obtained by mass genotyping, led to the identification of the line SDP035, derived from accession BGE003121, as a recombinant between the two gene pools, Andean and Mesoamerican [7]. The accession BGE04000 was identified as being of Andean origin, based on 12 molecular markers [35] and 3099 SNP markers [7].

In the present study, two approaches were used to improve the detection power and robustness of the GWAS: (i) analyses in multiple data sets and (ii) and multiple GWAS methodologies, based on both SL-GWAS and ML-GWAS. The control of false positives is very crucial in GWAS, but the effect of false negatives should not be ignored, which can occur if the cutoff value is too stringent. The low heritability observed for white mold resistance has led us to consider the less restrictive threshold value of −log(p) = 3. With these considerations, 22 WM QTNs were identified from the SDP. These QTNs were grouped into 15 QTIs: 3 QTIs on chromosome Pv01, 2 on Pv02, 2 on Pv03, 1 on Pv04, 4 on Pv08, and 3 on Pv09. No significant associations were identified on chromosomes Pv05, Pv06, or Pv07, on which other authors had described white mold QTLs [12,13,16,18,19,20,21,23]. It is important to note that different WM evaluation methods were used by different authors, so the results from different studies cannot always be compared. The seedling straw test under controlled conditions [39], as used in the present work, is the most frequently used method, but field evaluations [16,18,21,22] or a combination of both, controlled conditions and field, [12,13,17,19,23,40] were also used. Under field evaluations, there is a greater environmental influence and it can be difficult to distinguish between physiological resistance or avoidance mechanisms, such as growth habit, early development, the arrangement of leaves, or internode lengths.

Seven of the 15 QTIs described in the present research were identified for the first time at chromosomes Pv01 (QTI1_42), Pv02 (QTI2_34), Pv03 (QTI3_0.9), Pv04 (QTI4_2), Pv08 (QTI8_52, QTI8_55), and Pv09 (QTI9_36). The QTI4_2 was located at 2.02–2.47 Mbp on chromosome Pv04. It is not surprising that this chromosome region is involved in the resistant response against WM because it corresponds to a well-known cluster of resistance genes encoding protein kinases or proteins with NBS-LRR domains, which are typically involved in resistance response against pathogens [24]. In fact, resistance genes against multiple fungal diseases have been mapped at this position [47,48,49]. One of these genes is *Co-3*, which is a cluster of closely linked genes that confers resistance against *Colletotrichum lindemuthianum*, the causal agent of anthracnose of common bean. *Co-3* has been identified mainly in the Mesoamerican gene pool. This result could explain why Pv04 is identified in the WM response only in GWAS, where the panels include materials from the Mesoamerican gene pool, whereas most of the resistant sources used in bi-parental populations are of Andean origin [23].

The remaining eight QTIs identified in this work co-localized to WM QTLs, meta-QTLs or QTNs previously reported [23,34]:

The QTI1_45 (45.36–45.69 Mbp) corresponds to several QTLs identified from different populations (WM1.1^XC^, WM1.1^AO^, WM1.1^O83^) which together extends from 45.15 to 50.22 Mbp of chromosome Pv01 [23]. A chromosome region comprising 43.71–49.94 Mbp of Pv01 was also significantly associated with WM response in the GWAS [34]. QTI1_45 could be related to morphological plant traits involved in pathogen avoidance response, because at this position (45.56 Mbp) the gene *PHAVU_001G189200g* has been proposed as a candidate of *fin*, the gene controlling growth habit with recessive genotypes showing determinacy [50,51]. The implication of this chromosome region in avoidance mechanisms have also been suggested by other authors [12,20].

The QTI1_50 (50.97–51.17 Mbp of Pv01) is located close to the meta-QTL WM1.1 (49.57–50.22 Mbp) define by [23] and could correspond to the same region.

Correspondence was observed between QTI2_4 (4.53–5.68 Mbp), the meta-QTL WM2.2 (3.57–5.24 Mbp), and QTLs WM2.2^AN^ and WM2.2^R31^, which together extends from 3.91 to 21.76 Mbp of Pv02 [23]. A chromosome region between 3.14 and 4.61 Mbp of Pv02 was significantly associated with WM response in the GWAS [34].

The QTI3_39 (39.85–40.64 Mbp) co-localized with the meta-QTL WM3.1 (34.96–39.49 Mbp), with two QTLs WM3.1^AP^, WM3.1^XC^ (34.33–48.32 Mbp) [23], and the region between 33.55–44.48 Mbp of Pv03 was also identified in the GWAS [34].

QTI8_13 (13.22–13.42 Mbp) and QTI8_37 (37.04–37.24 Mbp) overlap with QTLs WM8.3^PS02−029C^ and WM8.3^Z0725^, which extend over most of chromosome Pv08, from 4.97 to 46.71 Mbp [23]. These regions (11.16–14.70 Mbp and 32.48–37.24 Mbp of Pv08) had also been identified in the GWAS [34]. Moreover, QTI8_37 is located less than 100 kbp from the meta-QTL WM8.3 (37.32–46.73 Mbp) so they could be considered the same region.

QTI9_11 (11.61–11.88 Mbp) and QTI9_16 (16.68–16.88 Mbp) extended from 8.19 to 17.73 Mbp, a region significantly associated with WM response in previous GWAS [34].

Among the genes underlying these QTIs, 61 were proposed as potential candidate genes, based on their function related to the three main stages proposed against the defense process: recognition, signal transduction, and defense response [11]. Among them, 15 Cytochrome P450 proteins were identified organized into two main clusters at QTI3_0.9 and QTI4_2, and 15 genes encoding typical R products like protein kinases or NBS-LRR proteins at QTI1_42, QTI1_45, QTI1_50, QTI2_4, QTI3_39, QTI4_2, QTI8_52, QTI9_11, and QTI9_16. Cytochromes P450s are a superfamily of enzymes involved in several plant functions, including response to biotic stresses [52]. For instance, transgenic plants of *Nicotiana benthamiana* overexpressing the gene *GmCYP82A3* of the cytochrome P450 family *CYP82* exhibited high resistance to gray mold (*Botrytis cinerea*) and black shank (*Phytophthora nicotianae*) [53].

Protein kinase and NBS-LRR genes constitute the largest plant disease resistance gene (R gene) family. Protein kinases are involved in defense mechanisms at different cellular levels, including elicitor recognition, as extracellular or intracellular receptors, in signal transduction, and the induction of transcriptional activation [54]. Furthermore, many NBS-LRR proteins recognize effectors secreted by pathogens directly or indirectly, which, in turn, activate downstream signaling pathways, leading to the activation of plant defense responses against various classes of pathogens, including bacterial, fungal, and viral, as well as nematode and insect pests [47,55]. Among the potential candidate genes identified, 28 encode for typical R genes products like protein kinases or NBS-LRR proteins and are located in all QTIs except for QTI8_55 and QTI9_36.

Genes *Phvul.002G055700* and *Phvul.002G055800*, both ethylene-responsive transcription factors, have been proposed as candidate resistance genes against WM by Vasconcellos et al. [23]. Ethylene signaling has been shown to be involved in defense against *S. sclerotiorum* in *Arabidopsis thaliana* [56] and the closely related *Brassica napus* [57]. Both genes, *Phvul.002G055700* and *Phvul.002G055800*, were identified in the present research at QTI2_4 and have been proposed as potential candidate genes against WM [23].

White mold produces many molecules and proteins, such as polygalacturonases (PGs), during infection and colonization to breach host cell walls [58]. Host plants produce polygalacturonase-inhibiting proteins (PGIPs) that recognize PGs and prevent enzymatic action during the invasion. In common bean, PGIPs consist of a complex locus of four genes organized in a cluster (PvPGIP1, PvPGIP2, PvPGIP3, and PvPGIP4) that spans a 50-kbp region at around 36 Mbp of chromosome Pv02 [59]. It has been demonstrated that this locus in the common bean is involved in defense against insects and fungi [59], including *S. sclerotiorum* infection [60,61].

Close to this PGIP cluster was located the QTI2_34 (34.54–34.84 Mbp), in which no candidate genes for WM resistance had previously been identified. It cannot be discarded that this cluster of PGIP genes may be the causal genes for WM response at this QTI. It is well known that SNPs may occasionally affect distant genes and the common practice of identifying candidate genes based on the nearest SNP association may lead to some difficulties in the identification of candidate genes [62,63].

## 5. Conclusions

The evaluation of the response to a local isolate of *S. sclerotiorum* in the SDP revealed 28 lines with high levels of resistance against this pathogen. Among these potential resistance sources, six were of Mesoamerican origin. This is a significant finding because most of the WM resistance sources identified to date are of Andean origin or from the secondary gene pool. The GWAS led to the validation of eight regions previously reported as being associated with WM resistance at chromosomes Pv01 (QTI1_45, QTI1_50), Pv02 (QTI2_4), Pv03 (QTI3_39), Pv08 (QTI8_13 and QTI8_37), and Pv09 (QTI9_11, QTI9_16). The GWAS also identified seven novel regions as being involved in the WM response located at chromosomes Pv01 (QTI1_42), Pv02 (QTI2_34), Pv03 (QTI3_0.9), Pv04 (QTI4_2), Pv08 (QTI8_52, QTI8_55), and Pv09 (QTI9_36). These results verify the complexity of this quantitative resistance, validate the implication of eight chromosome regions in WM response, and provide new targets that could be of major interest to bean breeders.

## Figures and Tables

**Figure 1 genes-11-01496-f001:**
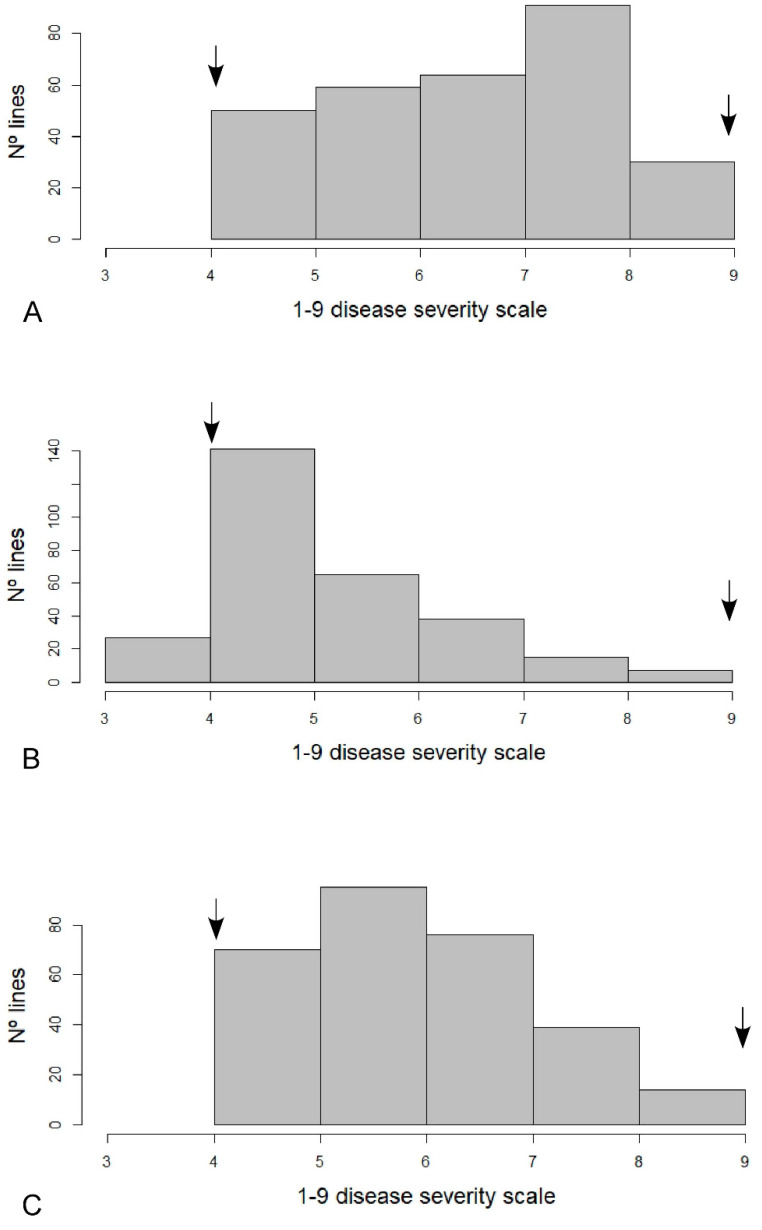
Histogram showing the distribution of the reaction against a local *Sclerotinia sclerotiorum* isolate in the 294 lines of the SDP (Spanish Diversity Panel) and the (**A**) E1, (**B**) E2, (**C**) Em. Arrows indicate the average value in the disease severity scale of resistant and susceptible checks.

**Figure 2 genes-11-01496-f002:**
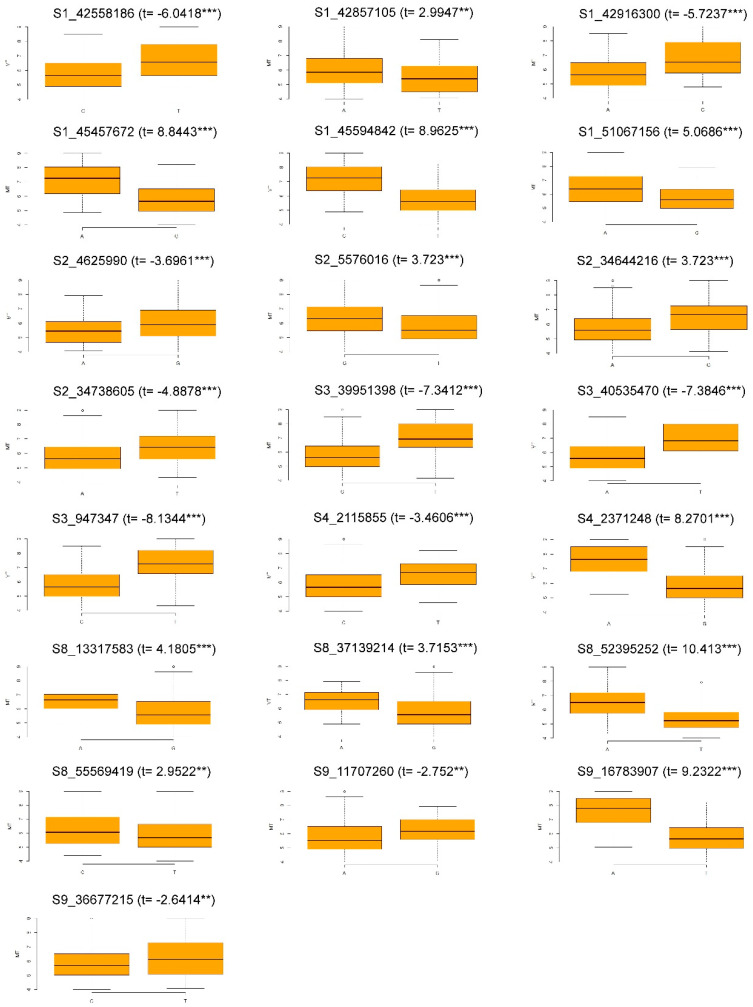
Box plot for the mean WM evaluation plotted according to the haplotype for the 22 SNPs associated with WM response. The X-axis represents the two alleles for each SNP, while the Y-axis corresponds to the phenotype (1 to 9 disease severity scale). T, T-student test value; ** *p* < 0.01; *** *p* < 0.001.

**Figure 3 genes-11-01496-f003:**
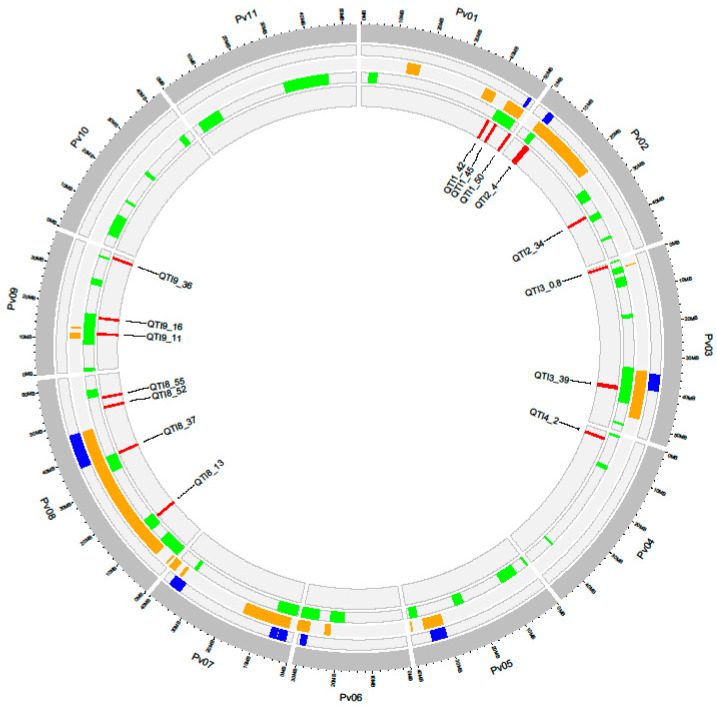
Circos plot showing the overlapping between WM quantitative trait intervals (QTIs) identified in this work (red track) and WM meta-quantitative trait loci (QTLs) (dark blue track), QTLs (orange track), and quantitative trait nucleotides (QTNs) (green track) previously described.

**Table 1 genes-11-01496-t001:** Lines included in the SDP (Spanish Diversity Panel) showing partial resistance against the *Sclerotinia sclerotiorum* local isolate. Resistant and susceptible checks are included. The severity scale value assigned in each evaluation (E1, E2, and Em) is indicated. According to Campa et al. [7], the accession/variety from which each SDP line was obtained, the gene pool (GP) assigned to each line (A, Andean; MA, Mesoamerican; M, recombinant between A and MA) and the growth habit (GH: det, determinate vs. ind, indeterminate) are indicated.

SDP	Accession/Variety	Type of Material	GP	GH	E1	E2	Em
SDP194	BGE043359	Spanish landrace	A	Ind	4.0	4.0	4.0
SDP217	CN_227	Spanish landrace	M	Ind	4.0	4.0	4.0
SDP289	Sacha	Commercial variety	MA	Ind	4.2	4.0	4.1
SDP203	BILMA	Commercial variety	MA	Ind	4.3	4.0	4.1
SDP216	CN_226	Spanish landrace	M	Ind	4.3	4.0	4.1
SDP246	Helda	Commercial variety	MA	Ind	4.1	4.1	4.1
SDP274	PLANETA	Commercial variety	MA	Ind	4.1	4.3	4.2
SDP226	DONNA	Commercial variety	MA	Ind	4.3	4.2	4.2
SDP108	BGE011762	Spanish landrace	A	Ind	4.3	4.3	4.3
SDP039	BGE003139	Spanish landrace	MA	Ind	4.2	4.4	4.3
SDP133	BGE022831	Spanish landrace	MA	Ind	4.7	4.0	4.3
SDP305	VITALIS	Commercial variety	MA	Ind	4.5	4.1	4.3
SDP234	FLORENCIA	Commercial variety	MA	Ind	4.4	4.3	4.4
SDP304	V381	Spanish landrace	MA	Ind	4.2	4.5	4.4
SDP106	BGE011736	Spanish landrace	M	Det	4.8	4.0	4.4
SDP107	BGE011758	Spanish landrace	MA	Ind	4.6	3.7	4.4
SDP255	MARCONI	Commercial variety	MA	Ind	4.6	4.2	4.4
SDP189	BGE039982	Spanish landrace	M	Ind	4.6	4.1	4.4
SDP025	BGE002196	Spanish landrace	M	Ind	4.4	4.5	4.5
SDP035	BGE003121	Spanish landrace	M	Ind	4.5	4.5	4.5
SDP053	BGE003482	Spanish landrace	A	Ind	4.4	4.6	4.5
SDP071	BGE004000	Spanish landrace	A	Ind	5.0	4.0	4.5
SDP088	BGE005475	Spanish landrace	MA	Ind	5.2	3.8	4.5
SDP221	CN_241	Spanish landrace	A	Det	4.6	4.4	4.5
SDP295	Tendergreen	Commercial variety	A	Det	5.0	4.1	4.5
SDP110	BGE013962	Spanish landrace	M	Ind	4.9	4.0	4.5
SDP150	BGE025745	Spanish landrace	A	Ind	5.0	4.1	4.5
SDP024	BGE002189	Spanish landrace	MA	Ind	5.0	4.0	4.5
	AB136	Resistant check	MA	Ind	4.0	4.0	4.0
	Cornell49242	Susceptible check	MA	Ind	8.9	9.0	9.0

**Table 2 genes-11-01496-t002:** Significant white mold (WM)-SNP associations identified using one single locus-genome-wide association study (SL-GWAS) method (mixed linear model—MLM), 6 ML-GWAS methods (mrMLM, FASTmrMLM, FASTmrEMMA, ISIS EM-BLASSO, pLARmEB, and pKWmEB), and three data sets (E1, E2, and Em). SNPs are named according to chromosome number and physical position in bp.

			E1	E2	Em
QTN	SNP	GWAS Method	−log_10_(*p*)	−log_10_(*p*)	−log_10_(*p*)
WM1_42.55	s1_42558186	MLM	3	-	3
WM1_42.85	s1_42857105	MLM	4	-	3
		pLARmEB	7	-	6
WM1_42.91	s1_42916300	FASTmrEMMA	7	-	5
WM1_45.45	s1_45457672	MLM	-	4	3
WM1_45.59	s1_45594842	MLM	-	6	4
		mrMLM	-	7	5
		FASTmrMLM	-	9	10
		ISIS EM-BLASSO	-	11	11
		FASTmrEMMA	-	7	5
		pLARmEB	-	6	8
		pLARmEB	5	-	7
WM1_51.06	s1_51067156	ISIS EM-BLASSO	3	-	3
WM2_4.62	s2_4625990	MLM	9	-	5
WM2_5.57	s2_5576016	pLARmEB	3	-	3
WM2_34.64	s2_34644216	MLM	-	5	4
WM2_34.73	s2_34738605	mrMLM	3	-	3
WM3_0.9	s3_947347	FASTmrMLM	-	4	5
WM3_39.95	s3_39951398	MLM	7	-	7
		pLARmEB	3	-	3
WM3_40.53	s3_40535470	MLM	-	6	5
WM4_2.11	s4_2115855	ISIS EM-BLASSO	-	4	3
WM4_2.37	s4_2371248	MLM	-	7	6
		pKWmEB	-	4	8
		pLARmEB	4	3	5
WM8_13.31	s8_13317583	MLM	-	3	3
WM8_37.13	s8_37139214	MLM	4	-	4
WM8_52.39	s8_52395252	MLM	7	-	7
		FASTmrMLM	6	6	7
		ISIS EM-BLASSO	10	-	10
		FASTmrEMMA	-	5	5
		pKWmEB	5	-	4
		pLARmEB	-	7	6
WM8_55.56	s8_55569419	pKWmEB	3	-	3
WM9_11.70	s9_11707260	MLM	-	5	4
WM9_16.78	s9_16783907	mrMLM	-	8	8
WM9_36.67	s9_36677215	pLARmEB	3	-	3

**Table 3 genes-11-01496-t003:** QTIs identified in the SDP for resistance to WM. The physical position of each QTI is indicated (chromosome, start, and end position in Mbp). The number of annotated genes and candidate genes underlying each QTI is indicated. Potential candidate genes were grouped in the main three stages described for WM response: recognition, signaling, and defense.

QTI	Chr	Start-End	QTNs	Nº Annotated Genes	Nº Candidate Genes	Stage of Resistance Response (Number of Genes)
QTI1_42	Pv01	42.46−43.02	WM1_42.55, WM11_42.85, WM1_42.91	46	8	*Recognition (6), Signaling (2)*
QTI1_45	Pv01	45.36−45.69	WM1_45.45, WM1_45.59	36	4	*Recognition (1), Signaling (1), Defense (2)*
QTI1_50	Pv01	50.97–51.17	WM1_51.06	28	2	*Recognition (1), Defense (1)*
QTI2_4	Pv02	4.53–5.68	WM2_4.62, WM2_5.57	74	7	*Recognition (3), Signaling (2), Defense (2)*
QTI2_34	Pv02	34.54–34.84	WM2_34.64, WM2_34.73	23	-	*-*
QTI3_0.9	Pv03	0.80–1.05	WM3_0.9	29	6	*Recognition (1), Defense (5)*
QTI3_39	Pv03	39.85–40.64	WM3_39.95, WM3_40.53	77	7	*Recognition (2), Signaling (1), Defense (4)*
QTI4_2	Pv04	2.02–2.47	WM4_2.11, WM4_2.37	42	12	*Recognition (1), Defense (11)*
QTI8_13	Pv08	13.22–13.42	WM8_13.31	6	-	*-*
QTI8_37	Pv08	37.04–37.24	WM8_37.13	2	-	*-*
QTI8_52	Pv08	52.30–52.50	WM8_52.39	24	5	*Recognition (4), Defense (1)*
QTI8_55	Pv08	55.47–55.67	WM8_55.56	22	1	*Signaling (1)*
QTI9_11	Pv09	11.61–11.88	WM9_11.70	24	2	*Recognition (2)*
QTI9_16	Pv09	16.68–16.88	WM9_16.78	15	3	*Recognition (1), Defense (2)*
QTI9_36	Pv09	36.58–36.78	WM9_36.67	20	4	*Signaling (1), Defense (3)*
				*468*	*61*

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
