# Peer review of "Genome-Wide Association Study (GWAS) for Resistance to Sclerotinia sclerotiorum in Common Bean"

_genes, 2020, doi:10.3390/genes11121496_

Round 1
Reviewer 1 Report
This is an interesting study showing results about the association between SNP markers and white mold (WM), an important fungal disease in common bean. Novel genomic regions associated with resistance reaction has been describe. In general, the manuscript is clear and well written besides, tables and figures (including supplementary files) supply a clear explanation of results. Methodology is appropriate, outstanding the careful design for disease resistance evaluation that has been done under controlled conditions. Foot notes in tables and figures have a very complete and comprehensive information.
The use of different GWAs methodologies to identify QTNs guarantee the reliability of the results. Authors provide a complete and informative discussion of results. This study gives novel information about plant material with high levels of resistance. Besides, inform about possible candidate genes involved in WM resistant reaction to be deeper studied in future.
I suggest minor changes
Line 132. Authors should include a reference to “Table 1”
Line 134: please include the meaning of the acronym SNAP (Supplemental Nutrition Assistance Program?)
Line 168: authors indicate: “Values equal to or less than 4.5 were considered to be a resistant reaction (R), values between 4.5 and 7 were considered to be an intermediate reaction (I), and values equal to or greater than 7 were considered to be susceptible (S).” Information about average values of susceptible and resistant checks should be included in the text.
Figure 1: It is recommendable to include arrows showing the average value of susceptible/ resistant checks

Author Response
This is an interesting study showing results about the association between SNP markers and white mold (WM), an important fungal disease in common bean. Novel genomic regions associated with resistance reaction has been describe. In general, the manuscript is clear and well written besides, tables and figures (including supplementary files) supply a clear explanation of results. Methodology is appropriate, outstanding the careful design for disease resistance evaluation that has been done under controlled conditions. Foot notes in tables and figures have a very complete and comprehensive information.
The use of different GWAs methodologies to identify QTNs guarantee the reliability of the results. Authors provide a complete and informative discussion of results. This study gives novel information about plant material with high levels of resistance. Besides, inform about possible candidate genes involved in WM resistant reaction to be deeper studied in future.
I suggest minor changes
Line 132. Authors should include a reference to “Table 1”
Response: reference number 7 (Campa et al. 2018) has been included in Table 1
Line 134: please include the meaning of the acronym SNAP (Supplemental Nutrition Assistance Program?)
Response: snap word is use in beans to refer a form of consumption for pods, when pods are harvested before the seed development phase. It is explained in the Introduction, lines 57-58, page 4.
Line 168: authors indicate: “Values equal to or less than 4.5 were considered to be a resistant reaction (R), values between 4.5 and 7 were considered to be an intermediate reaction (I), and values equal to or greater than 7 were considered to be susceptible (S).” Information about average values of susceptible and resistant checks should be included in the text.
Response: average values concerning susceptible and resistant checks has been included in Table 1.
Figure 1: It is recommendable to include arrows showing the average value of susceptible/ resistant checks.
Response: Arrows have been included in Figure 1.
Reviewer 2 Report
I suggest error control analysis for false positive with either Bonferroni or Hochberg procedure on significant SNPs (-log10(p) for the most stringent and more reliable result.
How was the variant calling done? 9000 SNPs looks very low. I expect a higher diversity for a core collection and hence larger number of SNPs.
PCA was described in the methods to show population structure of the collection but not shown anywhere in the results or discussion.
Line 143: The top significant SNP, s1_50223721, with MAF = 0.05 is in contrary to your filtering criteria.
The authors need to make correction for typos and grammatical errors. Below are some of these:
Line 144: Re-write the sentence and “chrs” should be written as “chromosomes”.
Line 158: “for” not “fort”
Line 202: “an” SNP?
Author Response
I suggest error control analysis for false positive with either Bonferroni or Hochberg procedure on significant SNPs (-log10(p) for the most stringent and more reliable result.
Response: Different methods have been used to establish the threshold of significance in GWAS such as Bonferroni correction, False Discovery Rate or Q value (Storey and Tibshirani https://doi.org/10.1073/pnas.1530509100). Bonferroni correction is frequently applied in the single-locus GWAS methods, but not for multi-locus GWAS (Zhang et al. 2019, doi: 10.3389/fpls.2019.00100). Bonferroni is a stringent correction that can results in the exclusion of important loci, especially for traits showing low heritability as white mold resistance (H2=0.33). For these reasons Bonferroni adjustment was not applied in this work but to address error control analysis of false positives, two approaches were used: (i) consistency between multiple environments and (ii) repeatability among GWAS methodologies: single-locus GWAS and multi-locus. This strategy allows us the identification of regions overlapping to several meta QTLs, also identified in different mapping populations, indicating the reliability of this approach GWAS.
How was the variant calling done? 9000 SNPs looks very low. I expect a higher diversity for a core collection and hence larger number of SNPs.
Response: The raw data have been processed by 2 steps: deleting adapter sequence, subsequently removing the reads, of which the rate of low quality (quality value <=7 (E) is more than or equal to 50%.Then about 78.15G bases were generated (clean data) for all the pooling lanes. Then all reads were assigned to the individuals by the ambiguous barcodes and the specific recognition site (GWCC). The reads without the unique barcodes and the specific sequence were discarded.
PCA was described in the methods to show population structure of the collection but not shown anywhere in the results or discussion.
Response: PCA matrix is necessary for conduct SL-GWAS and ML-GWAS. Results concerning PCA matrix and population structure of the Spanish Diversity Panel are deeply discuss in Campa et al. (2018, doi:10.3390/genes9110518). This paper is widely cited in the manuscript but considering the reviewer suggestion, it has also cited in line 200 of the new version.
Line 143: The top significant SNP, s1_50223721, with MAF = 0.05 is in contrary to your filtering criteria.
Response: The filtering criteria was MAF value equal or grater than 0.05. This have been corrected in the new version, line 143.
The authors need to make correction for typos and grammatical errors. Below are some of these:
Line 144: Re-write the sentence and “chrs” should be written as “chromosomes”.
Response: “Chr”, “chrs” have been edited all over the manuscript.
Line 158: “for” not “fort”
Response: corrected in the new version.
Line 202: “an” SNP?
Response: corrected in the new version.
Reviewer 3 Report
This paper is well-written and if published would be of high value to the research community. However, I have found several methodological shortcomings in this paper. A major shortcoming being the extremely low -log10(p-val) threshold being used here. I have provided my detailed objection within the attached document. Moreover, authors need to elaborate on and provide rationale for methods such as heritability, phenotypic variation and QTI intervals. As much as I liked the presentation style and scientific relevance of this article, I hope to see a methodologically rigorous version of this article. Therefore, I recommend a reconsider and resubmit for this article for now. I would be happy to review the updated version, should the authors decide to resubmit.
Some additional comments can be found within the attached document.

Author Response
This paper is well-written and if published would be of high value to the research community. However, I have found several methodological shortcomings in this paper. A major shortcoming being the extremely low -log10(p-val) threshold being used here. I have provided my detailed objection within the attached document. Moreover, authors need to elaborate on and provide rationale for methods such as heritability, phenotypic variation and QTI intervals. As much as I liked the presentation style and scientific relevance of this article, I hope to see a methodologically rigorous version of this article. Therefore, I recommend a reconsider and resubmit for this article for now. I would be happy to review the updated version, should the authors decide to resubmit.
Some additional comments can be found within the attached document.
L179: Did you mean to say “visualized” instead of “calculated”? Also replace chrs with chr
Response: It was modified in the new version. According to the reviewer 2 comments, chr/s was modified by chromosome/s all over the document.
L180: Please clarify what do you mean by independent inoculations here? I am guessing rep-to-rep correlation within each experiment?
Response: Correlations were calculated between inoculation E1 and E2. It was better explained in the new document.
L182: Please specify the model (in the form of equation) used for calculating the heritability. This would help clarify whether you are referrring to ‘repeatability’ or a broad-H2 by line means taking into consideration the two experiments (or sowing dates).
Response: Broad-sense heritability was calculated according to the ration of genetic variability (σ2g) to phenotypic variability (σ2g + σ2e). This equation was included in the updated document.
L187: Can remove the qualified ‘such a’ so that the sentence read: …architecture of a trait as complex as white mold…”
Response: Modified in the new version.
L188: Please clarify what do you mean by different datasets here? Different sowing date experiments, perhaps?
Response: In lines 172-174 it is explained what we refer with different data sets: “The two independent evaluations (E1, E2) and the mean value (Em) were considered to be three sets of data”.
Also, you may want to elaborate on why do you assume these two approaches can directly address the hypothesis of genetic dissection for white mold resistance?
Response: Two approaches were used to test the consistency or robustness of the identified WM response- SNP associations identified. We assume the importance of develop more than one independent evaluation to reduce environmental variation, especially for a trait showing low heritability (H2= 0.33) . We used different GWAS methodologies, all of them equally valid. We assume that the identification of one trait-SNP association in different mathematical models provide robustness to the results.
L197: How many PCs were selected for population structure, and based on which criteria?
Response: Two principal components were selected based on previous results (Campa et al. 2018, doi:10.3390/genes9110518). In this work, the optimal structure population of the Spanish Diversity Panel was estimated to be two, corresponding to the main gene pools described for common bean, Andean and Mesoamerican. The first component accounted for 40% of the variance and second principal component accounted for only 4.9% of the variance (see Campa et al. 2018, doi:10.3390/genes9110518). It has been specified in the new version.
L201: A p-value threshold of 3 being used here is extremely low considering the high number of single marker tests i.e. 5395. For a moment, even if we consider a relatibely lose FDR threshold with 0.10/5395, the number that comes out is 4.7. In my view, considering the discovery nature of this study, absolutely minimum threshold of -log10 of p-value of 4 has to be considered. Without a stringent thresold this test would be of relatively little use for the research community. This being said, I would be open to hearing a reasonable justification behind these p-val thresholds backed by a solid theoretical and/or evidence-based reasoning.
Response: Controlling for false positives is very crucial in GWAS, but the effect of false negatives should not be ignored, which can occur if the cutoff value is too stringent. The low heritability observed for white mold resistance has led us to consider the less restrictive threshold value of -log(p)=3. This criterium
was , some of the QTIs identified for a -log(p)=3 are validations of regions previously identified in other works. For example, QTI_50 was identified with a -log(p)=3 in the ISIS EM BLASSO model and corresponds with the meta-QTL WM1.1, identified in different biparental populations. We also consider the option that all the information concerning the GWAS analysis is available for the reader or researcher interested, who has the personal option of consider each QTI identified sufficiently robust or not for their own judgment.
L216: What was the rationale behind 100kbp cutoff for QTIs? It seems to me that the Maghadom paper cited here has used LD based cutoffs? Are you sure the chromosome wide LD block size is roughly 100kbp for beans?
Response: Different strategies have been used to identified the candidate genes involved in the genetic control of a trait as of SNP-trait associations. Moghaddam et al. studied the degree of inter- and intrachromosomal linkage disequilibrium present in their panel but they use the strategy for candidate gene search centered on the 100-kb region surrounding a significant SNP. The use of LD has the disadvantage that it is not a stable value along all the chromosomes (see Campa et al. 2018, doi:10.3390/genes9110518).
L237: The overall phenotypic analysis is lacking depth. I would like to see phenotypic distributions of each experiments and the repeatabilities. By simply looking at the broad-sense heritability of 0.33 here I can sort of guess there is a huge E or GxE effect (and extremely low G component) but I would like to see the real variance numbers corresponding to these factors to understand the quality of the ohenotypes better.
Response: Phenotypic distributions of experiment 1, experiment 2 and the average data have been included in the Figure 1 of the new version.
Round 2
Reviewer 2 Report
The manuscript can be accepted for publication.
Author Response
Authors really appreciate all comments and suggestions. Thank you.
Reviewer 3 Report
The authors have attempted to address my comments in the updated version. By and large most of the concerns have been addressed. However, there are still a few areas which require some clarification from the authors, and accordingly improvements. My detailed comments can be found in the attached document. I am providing a couple of bullet points here as a summary:
1. It is unclear as to how the adjusted means (e.g. the blups/blues) were obtained here. The authors described the phenotypic measurements in details but failed to provide any information on how the genotype-level means were calculated and the model used here. Please provide additional details on the method in the text, and through equations (if needed).
2. Heritability: this part has to be improved. Please provide heritability of the individual experiments since you are directly plugging those numbers into the GWAS analysis. It is difficult to assess the quality of the measurements in the absence of repeatibility/heritability numbers for each experiment. Also provide the updated heritability equation and update the results/discussion section accordingly.

Author Response
The authors have attempted to address my comments in the updated version. By and large most of the concerns have been addressed. However, there are still a few areas which require some clarification from the authors, and accordingly improvements. My detailed comments can be found in the attached document. I am providing a couple of bullet points here as a summary:
- It is unclear as to how the adjusted means (e.g. the blups/blues) were obtained here. The authors described the phenotypic measurements in details but failed to provide any information on how the genotype-level means were calculated and the model used here. Please provide additional details on the method in the text, and through equations (if needed).
Response: This part was improved in the new version (lines 178-186). See responses to the comments of the attached pdf document for more detail.
- Heritability: this part has to be improved. Please provide heritability of the individual experiments since you are directly plugging those numbers into the GWAS analysis. It is difficult to assess the quality of the measurements in the absence of repeatibility/heritability numbers for each experiment. Also provide the updated heritability equation and update the results/discussion section accordingly.
Response: This part was improved in the new document (lines 196-202). See responses to the comments of the attached pdf document for more detail.
Responses to the comments of the attached pdf document:
L173: How were the genotype-level disease ratings obtained? In other words, how were the multiple ratings per pot adjusted for GWAS analysis? If you haven’t done so, then I suggest please calculate adjusted means (aka BLUP/BLUEs) for each experiment separately and combined (by including GxE term in your linear model) before conducting any subsequent analyses.
Response: For each genotype, disease ratings were calculated as the arithmetic mean. Means were adjusted identifying outlier values through the coefficient of variation. Coefficient of variations over 50% were not accepted. This explanation was included in the new version (lines 178-186)
L182: It is still unclear whether the H2 was calculated for each experiment separately (E1 and E2) or calculated by including two experiments with the GxE term. If later is the case, then the denominator in your heritability equation would be as following: variance(g) + variance(GxE)/ number of experiments + variance(e)/ num experiments * num replications in each rep.
Response: In the new version, Heritability was estimated for each data set separately. A further explanation including the equations was included in lines 196-202.
L189: since you are conducting a GWAS separately on each experiment, it would be helpful for the audience to see the heritability for each of these data sets. The heritability that you reported here (0.33) is not as helpful since you are not conducting any GWAS on the combined dataset i.e. combined blups from E1 and E2.
Response: Included in the new version.
L204: The rationale you provided for slightly “permissive” p-values seems reasonable to me. Please go ahead and add a sentence here to briefly provide a context for the readers.
Response: Modified in the new version (lines 389-393).
L240: I see that you have now provided the phenotypic distributions in the form of figures. However, do add either the phenotypic variance components and/or heritabilities associated with the individual datasets.
Response: Figure 1 was modified in the new version, including the estimation of H2 for each data set.